# Prolactin Regulates Ovine Ovarian Granulosa Cell Apoptosis by Affecting the Expression of *MAPK12* Gene

**DOI:** 10.3390/ijms241210269

**Published:** 2023-06-17

**Authors:** Ruochen Yang, Chunhui Duan, Shuo Zhang, Yueqin Liu, Yingjie Zhang

**Affiliations:** 1College of Animal Science and Technology, Hebei Agricultural University, Baoding 071000, China; yangruochen1110@126.com (R.Y.); duanchh211@126.com (C.D.); liuyueqin66@126.com (Y.L.); 2College of Animal Science and Technology, China Agricultural University, Beijing 100089, China; 17835424912@163.com

**Keywords:** PRL, follicle counts, GC apoptosis, steroid hormone, *L-PRLR*, *S-PRLR*, *MAPK12*

## Abstract

Prolactin (PRL) has been reported to influence reproductive performance and cell apoptosis. However, its mechanism remains unclear. Hence, in the present study, ovine ovarian granulosa cells (GCs) were used as a cell model to investigate the relationship between PRL concentration and GC apoptosis, as well as its possible mechanisms. We examined the relationship between serum PRL concentration and follicle counts in sexually mature ewes. GCs were isolated from adult ewes and treated with different concentrations of PRL, while 500 ng/mL PRL was selected as the high concentration of prolactin (HPC). Then, we applied the transcriptome sequencing (RNA-Seq) combined with a gene editing approach to explore the HPC contributing to cell apoptosis and steroid hormones. The apoptosis of GCs gradually increased at PRL concentrations above 20 ng/mL, while 500 ng/mL PRL significantly decreased the secretion of steroid hormones and the expression of *L-PRLR* and *S-PRLR*. The results indicated that PRL regulates GC development and steroid hormones mainly through the target gene *MAPK12*. The expression of *MAPK12* was increased after knocked-down *L-PRLR* and *S-PRLR*, while it decreased after overexpressed *L-PRLR* and *S-PRLR*. Cell apoptosis was inhibited and the secretion of steroid hormones increased after interfering with *MAPK12*, while the overexpression of *MAPK12* showed the opposite trend. Overall, the number of follicles gradually decreased with increasing PRL concentration. HPCs promoted apoptosis and inhibited steroid hormone secretion in GCs by upregulating *MAPK12* through reducing *L-PRLR* and *S-PRLR*.

## 1. Introduction

Prolactin (PRL) is an endocrine polypeptide hormone released by the pituitary and extrapituitary of animals that not only plays an important role in lactation and maternal behavior but also regulates reproductive processes [1], such as ovarian growth and follicular development [2]. Previous studies have shown that PRL can alter the apoptosis and secretion of steroid hormones in mammalian and ovine ovarian granulosa cells (GCs): PRL induced *Bcl-2* expression to prevent apoptosis [3], while treatment with 100 ng/mL PRL induced an estradiol (E_2_) reduction and an FSH-induced progesterone (P_4_) increase in GCs cocultured with oocytes [4]. Moreover, the expression of the PRL receptor (*PRLR*) was detected on the surface of GCs in female ovaries, suggesting that PRL can bind to *PRLR* on GCs and affect the function of GCs directly [5]. *PRLR* is one of the single transmembrane proteins that are part of the cytokine receptor superfamily and plays an important role in the PRL signal transduction cascade [6]. At present, most studies have focused on the relationship between *PRLR* and reproductive function [7]. Ruminants have two types of *PRLR*, long *PRLR* (*L-PRLR*) and short *PRLR* (*S-PRLR*) [8]. Both isoforms’ genes were found in prezygotic follicle GCs and luteal cells within the corpus luteum, whereas *L-PRLR* was found specifically in stromal cells surrounding primordial and primary follicles [9]. It has previously been observed that *L-PRLR* is more important than *S-PRLR* in the follicular development of ewes [10]. Grosdemoug et al. [11] found that the absence of PRL/PRLR-JAK/STAT signal transduction led to a decrease in *LHR*, *FSHR,* and *PRLR* expression and consequently resulted in the loss of the enzyme cascade required to produce sufficient levels of P_4_ needed to maintain pregnancy. The deficiency of enzymatic cascades promoted follicular atresia, inhibited the function of GC aromatase involved in FSH, and decreased the E_2_ level, leading to reduced luteal function [11]. Females with endocrine abnormalities also have lower E_2_ levels, which can lead to high PRL levels in the blood, creating hyperprolactinemia (HPRL), which causes insufficient ovarian function, overflowing milk, and in severe cases, affecting follicle growth [12]. At the same time, it is also believed that the apoptosis of GCs caused by HPRL reduces the production of steroid hormones in the ovary, which affects the development and maturation of follicles, leading to abnormal menstruation and ovulation disorders and infertility [13,14,15]. Although the effect of PRL on GCs has been partially studied, the molecular mechanism of the regulation of HPC on ovine ovarian GCs is still poorly understood.

In our study, we investigated the regulatory mechanism of HPC on apoptosis and steroid hormone (E_2_ and P_4_) secretion in ovine ovarian GCs using the RNA-Seq, knockdown (CRISPR/Cas9), overexpression, and interference techniques. These results provide some clues to the PRL regulation mechanisms in follicle development and ovulation.

## 2. Results

### 2.1. Regression Analysis

H-E staining results of an ovine ovary treated with P_4_ and FSH are shown in Figure 1A,B. Univariate regression analyses of PRL on the number of follicles are shown in Figure 1C. R^2^ represents the proportion of variance explained by PRL and random effects over the overall variance, and R^2^ greater than 0.66 indicates a good fit. When using the polynomial regression model to predict follicle counts, the R^2^ for single independent variable linear, quadratic, and cubic models was 0.709, 0.7704, and 0.7795. The influence of PRL on follicles is strongly seen, and as PRL concentration increased, the number of follicle counts gradually decreased.

### 2.2. Immunofluorescence Staining

Because the FSHR protein was specifically expressed in GCs, the distribution of GCs and the GCs expressing FSHR can be seen intuitively by labeling FSHR and the nucleus (Figure 2C). The red marker denotes the cells expressing FSHR (Figure 2A) and the blue marker denotes the DAPI-stained nuclei (Figure 2B).

### 2.3. Cell Proliferation and Apoptosis Assay

With the increase in PRL concentration, cell viability increased first and then decreased (Figure 2D). Relative to the 4 ng/mL group, the level of cell viability was decreased in the 100 ng/mL and 500 ng/mL groups (*p* < 0.05), while the cell viability was lower in the 500 ng/mL group (*p* < 0.05) than in the 100 ng/mL and 20 ng/mL groups, and no significant difference was observed between the 20 ng/mL and 100 ng/mL groups.

The dead cells, late withered, early withered, and living cells were distinguished by the detection results (Figure 2E–I). The apoptotic rate of GCs in 500 ng/mL group was higher than those in the control, 4, 20, and 100 ng/mL groups (*p* < 0.05), and the 100 ng/mL group had a higher apoptosis rate of GCs than the control and 20 ng/mL groups (*p* < 0.05). No differences were observed among the control, 4 ng/mL, and 20 ng/mL groups (Table 1).

### 2.4. Effects of HPC on the Expression of L-PRLR and S-PRLR and the Secretion of Steroid Hormones (E_2_ and P_4_)

The results of HPC on the expression of *L-PRLR* and *S-PRLR* and secretion of steroid hormones are shown in Figure 3. The expression of *L-PRLR* was significantly higher than *S-PRLR* in the C group (*p* < 0.05; Figure 3A), but there was no difference between *L-PRLR* and *S-PRLR* in the P group. Compared with the P group, the expression levels of *L-PRLR* (*p* < 0.01) and *S-PRLR* (*p* < 0.05) in the C group were high.

The concentrations of E_2_ and P_4_ (Figure 3B) were significantly higher than those in the P group (*p* < 0.01).

### 2.5. DEGs Analysis

After removing the low-quality reads containing N sequences and adaptor sequences, a total of 190,029,146 raw reads and 183,895,812 clean reads were generated, and the fundamental details regarding sequencing data are contained in Table 2.

The Volcano map (Figure 4A) and cluster heat map (Figure 4B) of the DEGs were generated to further investigate the differentially expressed genes. mRNAs were differentially expressed (174 downregulated and 125 upregulated) with *p* < 0.05 and |1og2 fold change| ≥ 1 in the control as compared to the P group. Among the 125 upregulated genes, *HSD3B1* and *STAR* were associated with steroid hormone secretion, *CDC20* was related to the cell cycle, while *PLCE1*, *GRIA3*, *IGSF10,* and *CACNA1C* were enriched with functions correlated with cell proliferation and migration (Table 3). *RGS2* was associated with the cell cycle, the *MAPK12* gene was involved in the PRL pathway, the *CTSK* and *MAPK12* genes were closely related to ovulation, and *MAPK12* and *NFATC4* were involved in cellular senescence belonging to 174 downregulated genes (Table 3).

### 2.6. GO and KEGG Pathway Enrichment Analysis

To obtain insight into the biological significance, the enrichment of DEGs in GO terms was investigated. As a result, 418 GO terms were significantly enriched for DEGs, consisting of BP, CC, and MF for which the numbers were 321, 41, and 55, respectively. Several GO terms were discovered to be considerably enriched for DEGs, as shown in Figure 4C. The GO annotation indicated that the DEGs were involved in many BPs, such as cell morphogenesis involved in differentiation, the circulatory system process, regulation of synapse organization, and the multicellular organismal reproductive process. The main functional groups of DEGs in CC are the cell body, membrane, and lipoprotein, and in MF they are chemokine activity, protein tyrosine kinase activity, and proteoglycan binding.

The pathway analysis was carried out according to the KEGG pathway database to identify the significantly enriched pathways in DEGs. After pathway enrichment analysis, the top 20 upregulated (D) and downregulated (E) pathways are shown in Figure 4. The findings revealed that the significant signaling pathways contained 34 pathways. These include, for example, complement and coagulation cascades, cellular senescence, the cAMP signaling pathway, chemokine signaling pathway, toll-like receptor signaling pathway (TLR), oocyte meiosis, and Phospholipase D signaling pathway. Among them, the cAMP signaling pathway is the main pathway of cell proliferation and TLRs are closely related to ovulation, GC apoptosis, and steroid hormone synthesis [16,17,18].

### 2.7. RT-qPCR Confirmation

*ADCY3*, *CYP11A1*, *HSD13B1*, *STAR*, and *MAPK12*, related to cell proliferation and steroid hormone synthesis, were randomly selected to validate the expression profiles obtained by RNA-Seq in the same RNA samples. The qPCR results confirmed the DEGs between the C and P groups, which was consistent with the RNA-Seq findings (Figure 5). As a result, RNA-Seq can provide precise and efficient results for mRNA differential expression investigations.

It was shown that the expression of *MAPK12* was significantly upregulated in the P group, and the differential gene *MAPK12* was significantly enriched in the cell senescence signaling pathway, PRL signaling pathway, and the TRLs pathway, which is closely related to ovarian function. The related gene network mediated by *MAPK12* is shown in Figure 4F.

### 2.8. Expression of Related Genes after Knockdown and Overexpression of L-PRLR and S-PRLR

The cell fluorescence of each group after knockdown is shown in Figure 6A. No green fluorescence was observed after transfection by L-PRLR-sgRNA2, L-PRLR-sgRNA3, S-PRLR-sgRNA2, and S-PRLR-sgRNA3, indicating that the transfection failed. RT-qPCR was performed for the transfected L-PRLR-sgRNA1 (Figure 3C) and S-PRLR-sgRNA1 (Figure 3D) groups. The expression of *L-PRLR* was significantly lower (*p* < 0.01) in the P-sg-L group compared to the P and P-458 groups, while in the P-sg-SL group, the expression of *S-PRLR* was significantly lower (*p* < 0.01) than the P and P-458 groups. The expression of *L-PRLR* and *S-PRLR* both in the P and P-458 groups had no significant difference, proving the success of knockdown.

Cell fluorescence of each infected group is shown in Figure 6B. RT-qPCR was performed for the infected groups and the results are shown in Figure 3C,D. The expression of *L-PRLR* (Figure 3C) in the P-10-L group was significantly higher than that in the P and P-10 groups (*p* < 0.01). The expression of *S-PRLR* (Figure 3D) in the P-10-S group was significantly higher than that in the P and P-10 groups (*p* < 0.01). The expressions of *L-PRLR* and *S-PRLR* in both the P and P-10 groups have no significant difference, proving the success of overexpression.

The expression of *MAPK12* in each group is shown in Figure 3E. The expression of *MAPK12* was significantly higher (*p* < 0.01) in the P-sg-L and P-sg- SL groups than that in the P and P-458 groups, while the expression of *MAPK12* in the P-sg-SL group was significantly higher than that in the P-sg-L group (*p* < 0.05). No differences were observed between the P and P-458 groups. The expression of *MAPK12* in the P-10-L group was significantly lower than that in the P group (*p* < 0.05) and P-10 group (*p* < 0.01). The expression of *MAPK12* was reduced in the P-10-S group compared with that of the P and P-10 groups (*p* < 0.01). There was no difference between the P and P-10 groups as well as p-10-L and P-10-S groups. 

### 2.9. The Expression of Related Genes and the Secretion of Hormones in HPC GCs after Interference of MAPK12

The green fluorescence is for GCs successfully transfected or infected. P-458: GCs with HPCs were transfected by empty plasmid (negative control group for knockdown); P-sg-L: GCs with HPCs were transfected by recombinant plasmid of knockdown *L-PRLR*; P-sg-SL: GCs with HPCs were transfected by recombinant plasmid of both knockdown *L-PRLR* and *S-PRLR*; P-10: GCs with HPCs were infected by empty vectors of overexpression (negative control group for overexpression); P-10-L: GCs with HPCs were infected by lentiviruses carrying overexpressed sequences of *L-PRLR*; P-10-S: GCs with HPC were infected by lentiviruses carrying overexpressed sequences of *S-PRLR*; P-G: GCs with HPCs were infected by empty vectors of interference (negative control group for interference of *MAPK12*); P-SH2: GCs with HPCs were infected by lentiviruses carrying interfered sequence 2 of *MAPK12*; P-10-M: GCs with HPCs were infected by lentiviruses carrying overexpressed sequences of *MAPK12*.

Cell fluorescence of each infected group is shown in Figure 6C. The recombinant plasmid infection was successful in all three groups, and the results of RT-qPCR are shown in Figure 3F. The expression of *MAPK12* was lower in the P-SH1 and P-SH2 groups compared with the P (*p* < 0.01) and P-G (*p* < 0.05) groups. The expression of *MAPK12* in the P-SH3 group was lower than that in the P group (*p* < 0.05), while no difference was observed in *MAPK12* between the P-G and P-SH3 groups. The recombinant plasmid ShRNA2 of *MAPK12* was selected for the follow-up test.

#### 2.9.1. The Expression of Apoptosis-Related Genes

The expression of apoptosis-related genes (*Bcl-2*, *Bax,* and *Caspase3*) in HPC GCs after interference of *MAPK12* is shown in Figure 7A. The expression of pro-apoptotic gene *Caspase3* in the P-SH group declined significantly compared with the P and P-G groups (*p* < 0.01). The expression of pro-apoptotic gene *Bax* in the P-SH group was significantly lower than that in the P group (*p* < 0.05) and P-G group (*p* < 0.01), while the expression of anti-apoptotic gene *Bcl-2* increased in the P-SH group compared with the P and P-G groups (0.05 < *p* < 0.1). The results indicated that cell apoptosis could be inhibited by interference of *MAPK12* in HPC GCs.

#### 2.9.2. The Secretion of Steroid Hormones (E_2_ and P_4_) and PRL

The secretion of steroid hormones and PRL is shown in Figure 7C. The secretion of steroid hormones (E_2_ and P_4_) was significantly increased (*p* < 0.01) and the secretion of PRL was significantly decreased (*p* < 0.01) in the P-SH group compared with the P and P-G groups, while there were no differences in the secretion of E_2_, P_4_, and PRL between the P and P-G groups. These results indicated that steroid secretion could be promoted and PRL can be inhibited by the interference of *MAPK12* in HPC GCs.

### 2.10. The Expression of Related Genes and the Secretion of Hormones in HPC GCs after Overexpression of MAPK12

Cell fluorescence of each infected group is shown in Figure 6C and the results of RT-qPCR are shown in Figure 3F. The expression of *MAPK12* was significantly higher (*p* < 0.01) in the P-10-M group compared with the P and P-10 groups, which indicated that overexpression succeeded.

#### 2.10.1. The Expression of Apoptosis-Related Genes

The expression of apoptosis-related genes (*Bcl-2*, *Bax,* and *Caspase3*) in HPC GCs after overexpression of *MAPK12* is shown in Figure 7B. The expression of *Caspase3* in the P-10-M group has an increasing trend in comparison to the P and P-10 groups (0.05 < *p* < 0.1). The expression of *Bax* was higher in the P-10-M group than in the P-10 group (*p* < 0.05) and had an increasing trend in contrast to the P group (0.05 < *p* < 0.1). The expression of *Bcl-2* was lower in the P-10-M group than in the P-10 group (*p* < 0.05) and has a decreasing trend compared with the P group (0.05 < *p* < 0.1), indicating that cell apoptosis can be promoted in HPC GCs by overexpression of *MAPK12*.

#### 2.10.2. The Secretion of Steroid Hormones (E_2_ and P_4_) and PRL

The secretion of steroid hormones and PRL is shown in Figure 7D. The secretion of E_2_ and P_4_ was decreased significantly (*p* < 0.01) and the secretion of PRL was increased significantly (*p* < 0.01) in the P-10-M group. There were no differences in the secretion of E_2_, P_4_, and PRL between the P and P-10 groups. These results indicated that the steroid secretion can be inhibited and PRL can be promoted by overexpression of *MAPK12* in HPC GCs.

## 3. Discussion

### 3.1. The Effect of PRL on Proliferation and Apoptosis in GCs

The functions of PRL acting on cell proliferation and apoptosis as a growth factor by binding to *PRLR* have been purported for decades [3,16,18,19]. Previous work generally examined the relationship between PRL and cell proliferation and apoptosis. For example, in co-incubation, there was a C2-ceramide-induced increased apoptosis, which was inhibited in the presence of PRL [18], while the apoptosis rate of ovarian GCs increased significantly in the HPRL rat model [20]. However, the complicated mechanism of PRL-induced apoptosis in cells has not been studied in detail yet. In this study, the basic mechanism of PRL-induced apoptosis was characterized in GCs in vitro and Hu sheep follicle samples in vivo. We found that the viability of GCs was promoted in 4 ng/mL and 20 ng/mL groups and was inhibited in 100 ng/mL and 500 ng/mL groups, but the proliferation activity of GCs decreased significantly only when the concentration was 500 ng/mL, and this finding was consistent with previous study, which indicated that the proliferation of B lymphocytes and bovine osteoblasts was promoted at 10 ng/mL and 0.01 μg/mL–10.0 μg/mL of PRL, respectively [19]. Moreover, our results showed that with the increase in PRL concentration, the follicle count gradually decreased, and this finding was similar to that of previous study on rat ovaries [17]. The results suggest that PRL is likely to be extensively involved in the proliferation and apoptosis of ovarian GCs, which correlate with PRL concentrations.

### 3.2. The Effects of HPC on the Expression of L-PRLR and S-PRLR and the Secretion of Steroid Hormones (E_2_ and P_4_)

PRL operates by attaching to its receptor (*PRLR*) in target cells [8]. *L-PRLR* and *S-PRLR* appear to play distinctive functions in the ovary. For example, in mice, *L-PRLR* may be involved in folliculogenesis, while the role of *S-PRLR* is to form and maintain the corpus luteum [21]. The higher mRNA expression of *L-PRLR* in ovarian GCs, compared with *S-PRLR*, indicates that *L-PRLR* may be play a greater role than *S-PRLR* in follicular development [22]. Previous study has shown that the expression of *PRLR* in pregnant gilts with hyperprolactinemia has a downward trend [23]. In the present study, the expression of *L-PRLR* and *S-PRLR* was reduced by the high concentration of PRL, which is likely due to the HPC reducing the cascade reaction of PRL/PRLR, leading to a decrease in the number of *PRLRs* [24]. Previous study found that the absence of PRL/PRLR-JAK/STAT signal transduction led to a decrease in *LHR*, *FSHR,* and *PRLR* expression [25]. The diminished expression of *L-PRLR* and *S-PRLR* could lead to a lack of the enzymatic cascades, which are necessary to produce adequate levels of P_4_, and, in severe cases, complete cessation of P_4_ and reduced E_2_ levels, resulting in ovarian insufficiency [26]. Meanwhile, our study results showed a significant decrease in E_2_ and P_4_ in HPC GCs, which is consistent with the decrease in *PRLR*. A certain dose of PRL was needed to produce P_4_ during the ovarian follicular tissue culture of women, while the production could be inhibited with high dose of PRL [27]. Infertility and even amenorrhea are commonly observed in HPRL patients, which proves that a physiological dose of PRL can promote the synthesis of E_2_ and P_4_ [28] in the ovary, while the synthesis can be inhibited by a high concentration of PRL [16], which is in accordance with the results of the present study. Interestingly, previous studies have suggested that lacrimal gland (LG) cell proliferation is inhibited due to increased PRL levels and decreased serum levels of E_2_ and P_4_ in HPRL mice [16]. Therefore, the apoptosis of GCs caused by HPC in our experiment may be related to the decreased concentration of E_2_ and P_4_.

### 3.3. RNA-Seq Analysis

The cAMP signaling pathway is a major pathway that regulates cell proliferation [29]. Additionally, a previous study has suggested that TLR signaling has a role in the regulation of vertebrate ovarian function [30]. In this study, cellular senescence, the cAMP signaling pathway, oocyte meiosis, and the TLR pathway were the major enrichment pathways based on the KEGG pathway analysis, which was performed to predict the significantly enriched pathways in DEGs. Normal somatic cells withdraw from the cell cycle and demonstrate permanent growth arrest as a result of increasing telomere shortening in subsequent cell divisions, a condition known as cellular senescence. Lots of evidence suggests that stress-activated MAPK cascades that converge on JNK and c-Jun N-terminal kinase (JNKs) MAPKs have critical functions in the control of cellular senescence [31]. cAMP-PKA is the main pathway regulating GCs and the activation of cAMP in GCs leads to the activation of ERK1/2, ERK5, and P38MAPK cascade in the MAPK signal pathway, which is involved in the proliferation of GCs [32]. MAPK is triggered by a protein kinase phosphorylation cascade and is also involved in the regulation of meiosis in oocytes. As an intracellular signal transducer, cAMP regulates the activity of MAPK at different stages during meiosis in oocytes [33]. The MAPK pathway, which is associated with apoptosis and steroidogenesis of GCs, mainly consists of the extracellular receptor kinase (ERK) pathway, JNK, and P38 pathway [34,35]. One of the most common signaling pathways, the MAPK signaling cascade, is essential in cellular functions including differentiation, propagation, inflammation, apoptosis, and survival [36,37]. Estrogen also activates the MAPK pathway [38]. The present study showed that *MAPK12* was highly expressed in HPC GCs. Mitogen-activated protein kinases (MAPKs) are serine/threonine protein kinases that transform extracellular inputs into a range of cellular responses [39]. Wang et al. showed that MAPK plays a role in mediating the effect of TRH on PRL [40]. It has also been shown that intraperitoneal injection of PRL in female adult rats can rapidly and briefly activate MAPKs in the liver [41], which is consistent with the results of our study. *MAPK12* (*SAPK3*, *ERK6, or p38γ*) is one of a number of MAPKs that comprise four isoforms encoded [42] by different genes in mammalian cells, namely *MAPK14* (p38α), *MAPK11* (p38β), *MAPK12* (p38δ), and *MAPK13*. The protein sequences of these kinases are very comparable. The similarity between *MAPK14* and *MAPK11* was 75%, and the identity with *MAPK12* and *MAPK13* was 62% and 61%, respectively. Simultaneously, *MAPK12* and *MAPK13* have 70% similarity with each other [43]. The role in cell apoptosis, invasion, and migration of *MAPK12* has been proved through a new metabolomic approach [44]. Hence, HPC may activate the MAPK signaling pathway through different pathways to regulate the proliferation of GCs, and we selected *MAPK12*, which is the most differentially expressed gene in regulating ovarian GCs with HPC. However, the regulatory mechanism of HPC on apoptosis and steroid hormone secretion in sheep ovarian GCs has not been studied yet.

### 3.4. The Regulatory Mechanism of HPC on Apoptosis and Steroid Hormone (E_2_ and P_4_) Secretion in Sheep Ovarian GCs

The proliferation of GCs plays a crucial role in supporting the development of follicles and regulating the process of folliculogenesis, ultimately ensuring the production of mature and competent oocyte factors [16]. Emerging evidence indicates that PRL acts with their functions by interacting with *L-PRLR* and *S-PRLR* [8]. In the present study, the expression of *MAPK12* decreased after the knockdown of *L-PRLR* or *S-PRLR*, whereas the expression of *MAPK12* increased after overexpression of *L-PRLR* or *S-PRLR*, suggesting that HPC can promote apoptosis of GCs by upregulating *MAPK12* after reducing the expression of *L-PRLR* and *S-PRLR*.

The apoptosis of GCs is tightly regulated by a complex network involving multiple factors. Researchers explored the apoptosis mechanism of ovarian GCs at a molecular level and observed that the genes related to apoptosis of GCs can be roughly divided into two categories [45]: proapoptotic genes (*Bax*, *Caspase3*, etc.) and antiapoptotic genes (*Bcl-2*). *Bax* and *Bcl-2* belong to the same family and are also widely accepted as key factors of apoptosis [46]. The expression of genes in the Bcl-2 family is regulated by activated JNK and ERK, while the apoptosis of GCs regulated by the ERK pathway is mediated by inhibiting the activation of *Caspase3* directly or indirectly [47]. In this study, the expression of *Bcl-2* was upregulated, while *Bax* had the opposite trend in HPC GCs after the interference of *MAPK12*, in turn causing the inhibition of apoptosis in GCs. However, it has an opposite effect on Bax/Bcl-2 by overexpressing *MAPK12*. Meanwhile, in this study, the knockdown of *MAPK12* increased the secretion of steroid hormones, and, in contrast, overexpression treatment suppressed the secretion of steroid hormones. Ovarian GC steroids are closely related to cell proliferation and apoptosis [48] and are regulated by many signaling pathways, among which the MAPK pathway is one of the main pathways, with MAPKs being recognized as crucial targets for modulation [49]. Moreover, many growth hormones or cytokines regulate the secretion of steroid hormones by regulating the expression of related genes through the MAPK signaling pathway [50]. The results indicate that the influence of HPCs on GC apoptosis is mediated through the Bax/Bcl-2 pathway and the concentrations of E_2_ and P_4_, with regulation mediated by the *MAPK12* gene.

High concentrations of PRL inhibit the secretion of ovarian steroid hormones and cause luteolysis [51]. Previous study has shown that HPRL is an important cause of infertility, and hyperprolactinemic ovulation disorders account for 30% of infertile people [52]. At the same time, Xia et al. showed that a high concentration of PRL in sheep inhibited LH’s response to GnRH, thus causing ovulation to fail to start normally [53]. Therefore, the problems of ovulation abnormality and infertility caused by a high concentration of PRL need to be solved. Recent research has revealed that *MAPK14* plays a role in the production of prolactinoma, emphasizing the possibility of *MAPK14* as a potential targeted therapy in the treatment of prolactinoma [54]. Both *MAPK12* and *MAPK14* belong to the MAPK family genes, with 61% identity between them [43]. In this experiment, the secretion of PRL in HPC GCs with interference and overexpression of *MAPK12* was detected. It was found that PRL significantly decreased to a normal level after interference of *MAPK12*, while the result was the opposite after overexpression of *MAPK12*, suggesting that interference of *MAPK12* could reduce the secretion of PRL in HPC GCs, but the specific mechanism of these actions warrants further study. The findings provide fundamental evidence for the mechanism by which HPC regulates ovarian GC apoptosis, providing a basis for future gene editing interventions in addressing cases of ovulation failure in sheep [55,56] caused by high levels of PRL.

## 4. Materials and Methods

### 4.1. Animals

Sexually mature, healthy, second-parity Hu sheep (*n* = 24) with an initial body weight of 35.16 ± 3.66 kg were used as experimental animals. Twenty-four sheep with good estrus synchronization (estrus in 60 h; estrus Rate, 89%) were randomly allocated to one of four groups with the same dietary treatments. The animals were maintained in an open paddock and fed a total mixed ration (Appendix A), including 1.27 kg corn silage, 0.348 kg peanut seeding, and 0.775 kg concentrate mixture. The daily ration was divided into two equal portions and supplied to the sheep at 0700 h and 1700 h. The sheep had free access to drinking water. At the beginning of the trial, all sheep were induced to estrus synchronization according to the methods described previously [57]. The experimental design is shown in Appendix A.

### 4.2. Correlation between PRL and Follicle Counts

To evaluate the correlation between PRL concentration and follicle counts, twenty-four ewes were randomly divided into four groups (A, B, C, D) with six ewes in each group. Five days after removing the vaginal sponge, all ewes were treated with a progesterone (P_4_) vaginal sponge. The sponges were removed on the 8th day for group A and group C and removed on the 11th day for group B and group D. Group C received 100 IU (70 IU injected at 8:00 a.m., 30 IU injected at 6:00 p.m.) follicle-stimulating hormone (FSH) by intramuscular injection on day 7, and group D received the same dose of FSH as group C at the same time point on day 10; then, the blood was taken from the jugular vein into 5 mL vacuum tubes containing K_2_EDTA for all sheep before morning feeding on the 8th and 10th days, respectively, and immediately centrifuged at 3500 r/min for 15 min at 4 °C to obtain plasma samples, which were kept at −20 °C for further analysis of PRL. Meanwhile, unilateral ovaries of each ewe were collected to count the number of follicles.

The plasma PRL was analyzed using the enzyme-linked immunosorbent assay method, and the procedures for analysis were according to the manufacturer’s instructions using commercially available kits (Shanghai Enzyme Linked Biotechnology Co., Ltd., Shanghai, China; Sheep Prolactin ELISA Kit; NO. JLC10241; Sensitivity > 15 pg/mL). We counted the total number of small antral follicles (1–2 mm), medium antral follicles (2–4 mm), and large antral follicles (>4 mm) in the ovaries as previously described [58]. Ovaries were cut into successive 5 μm slices, the first 10 slices were first discarded, and then samples were collected every 10 slices, for a total of 5 slices per ovary. All the small, medium, and large antral follicles on each of the 5 sections were counted by image J after H-E staining.

### 4.3. Experiment 1: PRL Model and Target Gene Identification

#### 4.3.1. Ovaries GC Culture and Experimental Design

To further explore the possible mechanisms regarding how PRL regulates GC apoptosis and steroid hormone secretion, twenty-four sexually mature ewes aged between 1 and 1.5 years were reselected, and the fresh ovaries were collected, which were kept at 37 °C and immediately placed in a buffered saline solution supplemented with streptomycin/penicillin mixture (1%) and immediately transported to the laboratory within 4 h. After three rounds of sterile DPBS cleaning, dominant follicles with diameters ranging from 3 to 7 mm were isolated from the ovaries. Then, GCs were collected from the interior of the follicles with a 1 mL injector, and the GCs were seeded in six-well cell culture plates at a density of 1 × 10^5^ per well and cultured in a mixed culture medium that consisted of 89% DMEM/F12, 10% fetal bovine serum (FBS), and 1% streptomycin/penicillin mixture, with five levels of PRL (ovine PRL: PROSPEC, cyt-240, purity ≥99.0%), i.e., 0, 4, 20, 100, and 500 ng/mL were added to the medium as substrate and then incubated at 37 °C and 5% CO_2_ for 24 h. The GCs were identified by the immunofluorescence method as described previously [59].

#### 4.3.2. Cell Proliferation and Apoptosis Assay

GCs were seeded in 96-well culture plates at a density of 5 × 10^3^ per well and treated with 0, 4, 20, 100, or 500 ng/mL PRL for 24 h. Each well received 10 μL CCK-8, and the cells were hatched in the dark for 1–4 h. The optical density of the cells at 450 nm was measured using a microplate reader. In addition, each experiment was repeated four times.

GCs were inoculated into 6-well plates at a density of 1 × 10^5^ per well and incubated with 0, 4, 20, 100, or 500 ng/mL PRL for 24 h. The cells were washed with PBS and resuspended in a 1 Annexin V binding buffer. The resultant cultures were incubated in the dark for 15 min with Annexin V-FITC and PI reagent. Ultimately, the apoptosis was determined in 1 h using a BD FACSCanto II Flow Cytometer. In addition, each experiment was performed in triplicate.

#### 4.3.3. P_4_ and E_2_ Analysis

An amount of 1 mL of cell supernatant culture medium for each group was collected from the six-well plate, separated by centrifugation (laboratory centrifuge: SIGMA-3K15; 1000× *g*/10 min), and stored at −20 °C for subsequent analysis. Concentration of P_4_ (Shanghai Enzyme Linked Biotechnology Co., Ltd.; Sheep Progesterone ELISA Kit; NO. JLC10263; Sensitivity > 0.1 ng/mL) and E_2_ (Shanghai Enzyme Linked Biotechnology Co., Ltd.; Sheep Estrogen ELISA Kit; NO. JLC10385; Sensitivity > 1 pg/mL) was measured following the manufacturer’s instructions with the respective analytical kits. The intraassay CV was 10%.

#### 4.3.4. The GC Model of PRL

At present, the establishment of a PRL model mainly includes an estrogen-induced prolactinoma model and a transgenic animal prolactinoma model [60]. Neradugomma et al. investigated the mechanism of PRL-induced Notch signaling by adding 500 ng/mL of PRL to colon stem cells [61]. In the present study, we found that granulosa cells (GCs) treated with 500 ng/mL PRL showed a significantly lower concentration of P_4_ and cell viability than other groups. Therefore, to evaluate the influence of a high concentration of PRL (HPC) on ovarian GCs, and combine this with the results of previous studies, 500 ng/mL PRL was chosen as the PRL model for subsequent experiments. So, the C group (blank cells) and P group (500 ng/mL PRL) were established with 4 replicates in each group. The culture method and P_4_ and E_2_ analysis was the same as above.

#### 4.3.5. RNA Extraction, Library Preparation, and Sequencing

Ovine GCs subjected to the C groups (*n* = 4, 0 ng/mL PRL groups) and P groups (*n* = 4, 500 ng/mL PRL groups) were used for RNA-Seq. Total RNA of ovine GCs was extracted using Trizol (Invitrogen, Carlsbad, CA, USA) according to the manufacturer’s instructions. The RNA concentration, purity, and integrity were analyzed with a Qubit RNA Assay Kit (Life Technologies, Carlsbad, CA, USA), Nano photometer Spectrophotometer (Thermo Fisher Scientific, Waltham, MA, USA), and RNA Nano 6000 Assay Kit of the Bioanalyzer 2100 system (Agilent Technologies, Carlsbad, CA, USA), respectively. RNA (3 μg) of each sample with a 28S/18S ratio > 1.8 and an OD 260/280 ratio > 1.9 was used for library construction according to the IlluminaTruSeqTM RNA Kit protocol (Illumina, San Diego, CA, USA). The transcriptome sequencing was conducted by Novogene Bioinformatics Technology (Beijing, China).

#### 4.3.6. Bioinformatics Analysis of RNA-Seq

FastQC was used to assess the quality of the raw reads (v0.11.9). Trimmomatic (v0.39) was then used to eliminate low-quality reads, poly-N, and adaptor sequences from the reads. HISAT2 was utilized after quality control to match high-quality sequences produced here against the sheep reference genome and map them to genomic site positions [62]. Remaining gene counts were normalized using the Fragments Per Kilobase of transcript per Million (FPKM) mapped reads method to account for varying gene lengths and sequencing depth between samples. Gene expression levels calculated were used to assess gene expression discrepancies between samples.

#### 4.3.7. Differentially Expressed Genes and Target Gene Identification

Deseq was used to identify the differentially expressed genes (DEGs), using the (v1.18.0) package based on the read count data. Genes in the threshold of |FoldChange| ≥ 1.5 and false discovery rate (FDR) < 0.01 were assigned as significantly differentially expressed genes.

To preliminarily identify candidate genes for PRL-regulated GCs, the “clusterProfiler” program was used to conduct gene ontology (GO) enrichment analysis of differentially expressed genes from the biological process (BP), molecular function (MF), and cellular component (CC) [63,64]. DEGs were examined for pathway enrichment in the Kyoto Encyclopedia of Genes and Genomes (KEGG), and KEGG pathways adjusted for *p* values less than 0.05 were regarded as significantly enriched in DEGs. The relationship between genes and their associated GO keywords and pathways was then plotted using Cytoscape v3.8.0 [65]. The genes associated with GC activity and steroid hormone metabolic pathways should be highlighted and selected for subsequent analysis [64]. In the present study, a total of 299 differential genes were obtained; among the DEGs, the *MAPK12* gene is the most associated with cellular vitality and steroid hormone secretion-related pathways. Particularly, the *MAPK12* gene is involved in the PRL pathway, ovulation, and cellular senescence. Therefore, the *MAPK12* gene was selected as a target gene.

#### 4.3.8. GCs of HPC Were Treated with CRISPR/Cas9 and Overexpression Techniques

Knockdown by CRISPR/Cas9: Three 20–25 bp sgRNAs (small guide RNA) (Table 4) targeting *L-PRLR* and *S-PRLR* were designed (https://benchling.com/signin, accessed on 10 November 2021) and chemically synthesized to construct CRISPR expression vectors for transfection in ovine ovarian GCs. GCs with HPCs were transfected by a recombinant plasmid of knockdown *L-PRLR* (P-sg-L group), empty plasmid (negative control group: P-458 group, the empty plasmid px458 of knockdown is shown in Appendix A), recombinant plasmid of both knockdown *L-PRLR* and *S-PRLR* (P-sg-SL group), and HPC GCs (P group). Then, 1 × 10^5^ cells of each group were seeded into a 6-well plate to analyze the expression of *L-PRLR* and *S-PRLR*. After overnight incubation at 37 °C, cells were transfected with recombinant plasmids or Control Plasmid pX458 using Lipofectamine 3000 (Invitrogen, Carlsbad, CA, USA). Sequences for sgRNAs are shown in Table 4. Fluorescence was observed at 48 h and the samples were collected for RT-qPCR detection to verify whether the knockdown was successful.

Overexpression: All synthetic plasmids carrying the lentivirus were obtained from Jiangsu Genewiz Biotechnology Co., Ltd (Suzhou, China). GCs with HPC were infected by lentiviruses carrying overexpressed sequences (Appendix A) of *L-PRLR* (P-10-L group), lentiviruses carrying overexpressed sequences (Appendix A) of *S-PRLR* (P-10-S group), lentiviruses with empty vectors of overexpression (negative control group: P-10 group, the empty plasmid pGWLV10-new of overexpression is shown in Appendix A), and the P group. For overexpression, 1 × 10^5^ cells were added per well in a 6-well plate and were infected by the optimum multiplicity of infection (MOI = 400) of each group’s lentivirus at 50% confluence. Following 6 h of incubation, the lentivirus was replaced with fresh growth medium. The cells were taken 48 h after infection to check the overexpression effect.

### 4.4. Experiment 2: Functional Validation of MAPK12

#### 4.4.1. Interference and Overexpression of MAPK12 in GCs with HPC

All synthetic plasmids carrying the lentivirus were obtained from Jiangsu Genewiz Biotechnology Co., Ltd. GCs with HPC were infected by lentiviruses with empty vectors of interference (negative control group: P-G group, the empty plasmid pGWLV33 of interference is shown in Appendix A), lentiviruses carrying the interference sequence of *MAPK12* (P-SH group), lentiviruses carrying overexpressed sequences (Appendix A) of *MAPK12* (P-10-M group), the P-10 group, and the P group. The sequences of ShRNAs are shown in Table 4 and the infection method was the same as mentioned above.

#### 4.4.2. RNA Preparation and RT-qPCR

Total RNA was extracted according to the specification of the TRNzol total RNA extraction kit (TIANGEN, Beijing, China) and stored at −80 °C. Reverse transcription was carried out using a reagent kit (Takala, Beijing, China) according to the manufacturer’s instructions. The reaction mixture, 20 µL, consisted of 5× Mix (4 µL), RNA (2 µg), and RNase-free water (16 µL). The reaction was performed at 37 °C for 15 min followed by 85 °C for 5 s. The product was stored at −4 °C.

Primers of *ADCY3*, *CYP11A1*, *STAR*, *MAPK12*, *HSD13B1*, *L-PRLR*, *S-PRLR*, *MAPK12*, *Bax*, *Bcl-2*, *Caspase3*, and *GAPDH* were created using Primer Premier 5.0 from the conserved region and synthesized by Jiangsu Genewiz Biotechnology Co., Ltd. The primers used for RT-qPCR are listed in Table 5. RT-qPCR was carried out in strict accordance with the instructions for the LC-480 PCR system, using the Ultra SYBR Mixture (with Rox) and the following cycling protocol: 2 min at 94 °C, followed by 40 cycles of 5 s at 95 °C and 30 s at 60 °C. The reaction mixture volume was 20 μL, which contained 10 μL Ultra SYBR Mixture (2×), 0.4 μL upstream and downstream primers (10M), 2 μL template, and sterile distilled water. The 2^−△△CT^ approach was used to calculate relative expression levels based on quantitative real-time PCR results.

### 4.5. Statistical Analysis

Data from the hormone concentration, RT-PCR study, and relative expression of *L-PRLR* and *S-PRLR* between the C group and P group were analyzed using the *t*-test in SPSS software (ver. 22.0, IBM Corp., Armonk, NY, USA). The one-way ANOVA procedure in SPSS software followed by Duncan’s post hoc test was used to compare the relative expression of *L-PRLR*, *S-PRLR,* and *MAPK12* among multiple groups. *p* values were corrected for multiple comparisons at different groups using the Bonferroni correction. Differences were considered significant at *p* < 0.05. The normal distribution of all data was determined according to the Shapiro–Wilk normality test, and the homogeneity of the variance test was detected by a Bartlett test. The regression and correlation analysis between PRL and follicle count was analyzed using the “lm” package in R [66]. The results are expressed as mean ± Standard Error of the Mean (SEM). GraphPad Prism 9.0 software and the R language (R-3.5.2) “ggplot2” package were used for visualization mapping.

## 5. Conclusions

A low concentration of PRL inhibited apoptosis. A high concentration of PRL (HPC) promoted apoptosis and inhibited the secretion of steroid hormones (E_2_ and P_4_) of GCs in the ovary of ewes by reducing the expression of *L-PRLR* and *S-PRLR*, resulting in upregulating *MAPK12* expression.

## Figures and Tables

**Figure 1 ijms-24-10269-f001:**
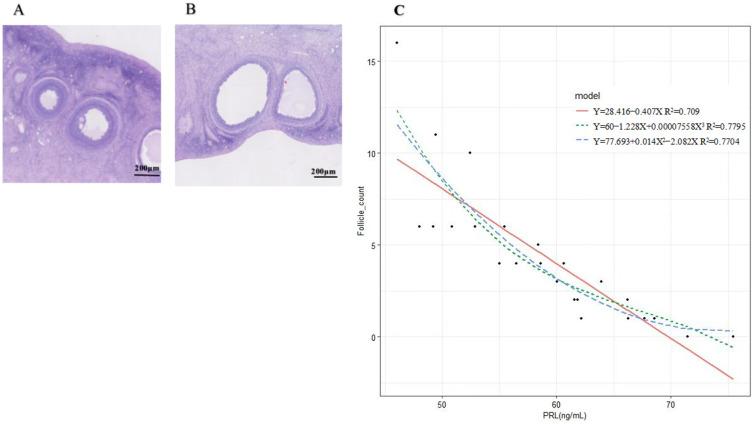
H-E staining and fitting curve of prolactin concentration and follicle number. (**A**,**B**): H-E staining results of ovine ovary treated with P_4_ and FSH; (**C**): Fitting curve of prolactin concentration and follicle number.

**Figure 2 ijms-24-10269-f002:**
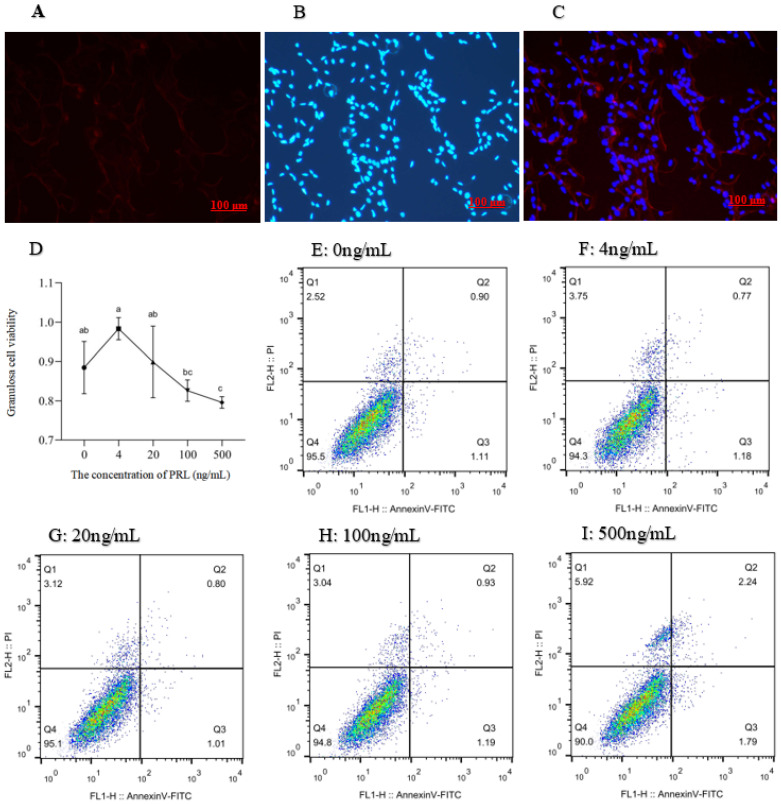
Identification of ovine ovarian granulosa cells and treatment with different concentrations of PRL on viability (450 nm OD) and apoptosis in cultured ovine ovarian granulosa cells. (**A**): The red marker denotes the cells expressing FSHR; (**B**): The blue marker denotes the DAPI-stained nuclei; (**C**): Merged are a red fluorescently labeled FSHR and a blue fluorescently labeled DAPI overlay; (**D**): The different lowercase letters indicate significant differences (*p* < 0.05); (**E**–**I**): The *X* axis represents PI fluorescence; *Y* axis represents Annexin-V fluorescence. Q1: The dead cells; Q2: The late withered; Q3: The early withered; Q4: The living cells.

**Figure 3 ijms-24-10269-f003:**
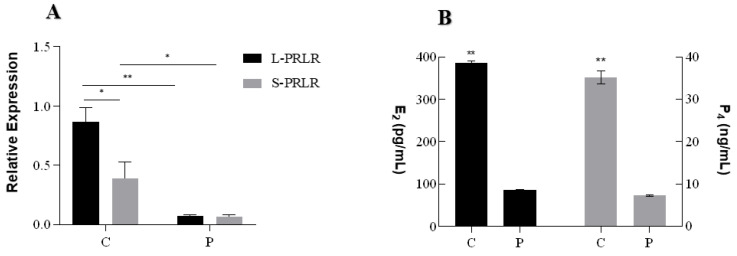
qPCR of *L-PRLR*, *S-PRLR,* and *MAPK12* and the secretion of steroid hormones (E_2_ and P_4_). (**A**): Relative expression of *L-PRLR* and *S-PRLR*; (**B**): The secretion of steroid hormones (E_2_ and P_4_); (**C**): qPCR validation of *L-PRLR* after knockdown and overexpression; (**D**): qPCR validation of *S-PRLR* after knockdown and overexpression; (**E**): The relative expression of *MAPK12* after knockdown and overexpression of *L-PRLR* and *S-PRLR*, respectively; (**F**): qPCR validation of *MAPK12* after interference and overexpression. “*” and “**” indicate 0.01 < *p* < 0.05 and *p* < 0.01, respectively. *L-PRLR*: Long Prolactin Receptor; *S-PRLR*: Short Prolactin Receptor; C: Control group (0 ng/mL PRL); P: GCs treated with 500 ng/mL PRL (GCs with HPCs); P-458: GCs with HPCs were transfected by empty plasmid (negative control group for knockdown); P-sg-L: GCs with HPCs were transfected by recombinant plasmid of knockdown *L-PRLR*; P-sg-SL: GCs with HPCs were transfected by recombinant plasmid of both knockdown *L-PRLR* and *S-PRLR*; P-10: GCs with HPCs were infected by empty vectors of overexpression (negative control group for overexpression); P-10-L: GCs with HPCs were infected by lentiviruses carrying overexpressed sequences of *L-PRLR*; P-10-S: GCs with HPC were infected by lentiviruses carrying overexpressed sequences of *S-PRLR*; P-G: GCs with HPCs were infected by empty vectors of interference (negative control group for interference of *MAPK12*); P-SH1: GCs with HPCs were infected by lentiviruses carrying interfered sequence 1 of *MAPK12*; P-SH2: GCs with HPCs were infected by lentiviruses carrying interfered sequence 2 of *MAPK12*; P-SH3: GCs with HPCs were infected by lentiviruses carrying interfered sequence 3 of *MAPK12*; P-10-M: GCs with HPCs were infected by lentiviruses carrying overexpressed sequences of *MAPK12*.

**Figure 4 ijms-24-10269-f004:**
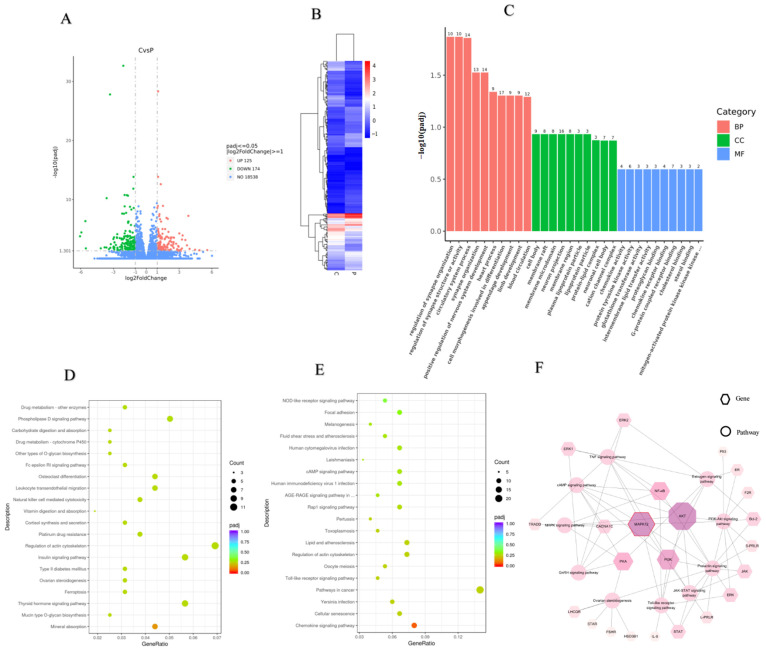
Analysis of the RNA-Seq. (**A**): Volcano map of differentially expressed genes between C and P groups; (**B**): Cluster heat map of differentially expressed genes between C and P groups; (**C**): GO function enrichment of differentially expressed genes (the numbers at the top of the bars) between C and P groups; (**D**): KEGG pathway analysis of upregulated DEGs of differentially expressed genes (top 20) between C and P groups; (**E**): KEGG pathway analysis of downregulated DEGs of differentially expressed genes (top 20) between C and P groups; (**F**): The network regulation of genes mediated by *MAPK12*. KEGG: Kyoto Encyclopedia of Genes and Genomes; DEGs: differentially expressed genes; BP: biological process; MF: molecular function; CC: cellular component. C: control group (0 ng/mL PRL); P: GCs treated with 500 ng/mL PRL (GCs with HPCs).

**Figure 5 ijms-24-10269-f005:**
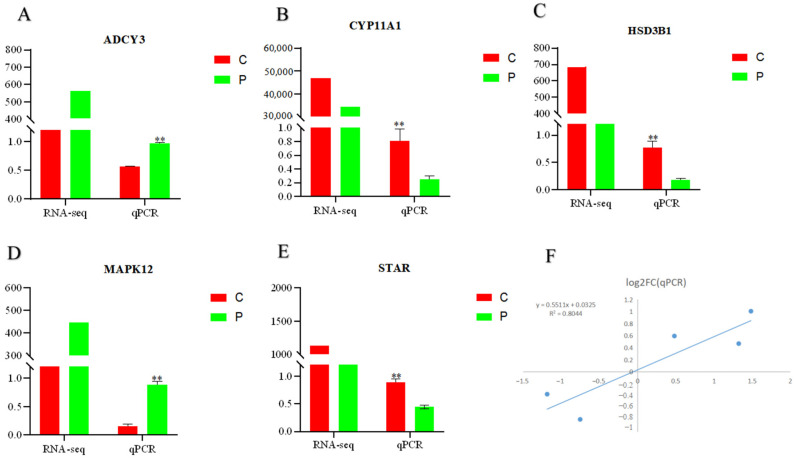
qPCR validation of RNA-Seq. (**A**–**E**): Five coding genes; (**F**): Correlation analysis between qPCR and RNA-Seq results. The *Y*-axis indicates the log_2_FC value according to qPCR and the *X*-axis indicates the log_2_FC value according to RNA-Seq. “**” indicate *p* < 0.01. (**C**): Control group (0 ng/mL PRL); P: GCs treated with 500 ng/mL PRL (GCs with HPCs).

**Figure 6 ijms-24-10269-f006:**
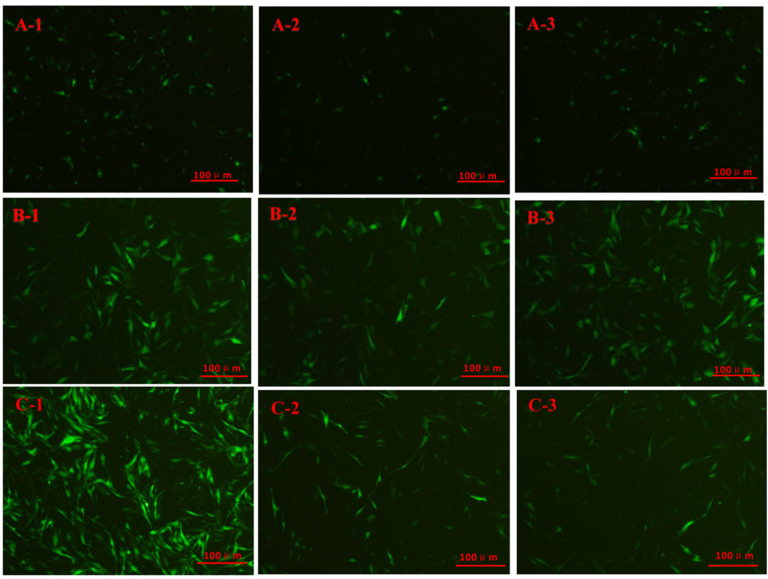
Fluorescence of each group of cells. (**A-1**): P-458 group; (**A-2**): P-sg-L group; (**A-3**): P-sg-SL group; (**B-1**): P-10 group; (**B-2**): P-10-L group; (**B-3**): P-10-S group; (**C-1**): P-G group; (**C-2**): P-SH2 group; (**C-3**): P-10-M group.

**Figure 7 ijms-24-10269-f007:**
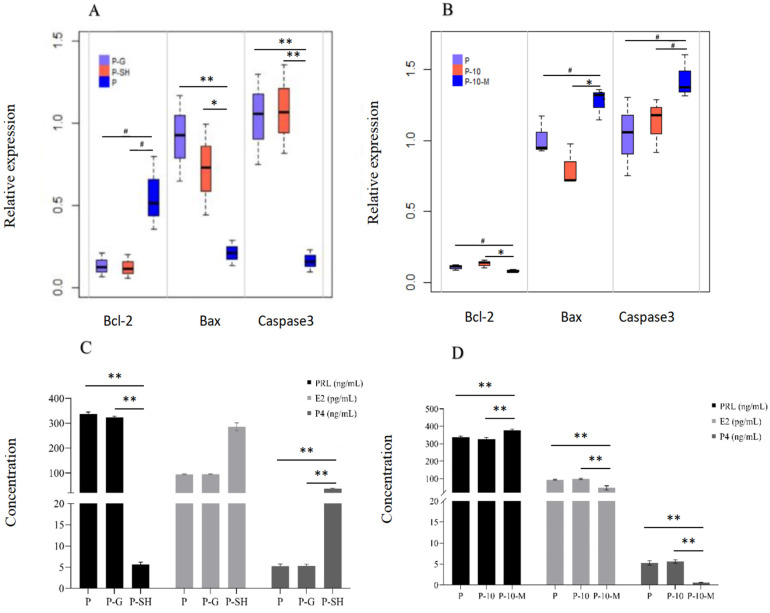
Expression of apoptosis-related genes and the secretion of steroid hormones and PRL after interference and overexpression of *MAPK12* in cells of a high concentration of PRL. (**A**): Expression of apoptosis-related genes after interference of *MAPK12* in cells of a high concentration of PRL; (**B**): Expression of apoptosis-related genes after overexpression of *MAPK12* in cells of a high concentration of PRL; (**C**): The secretion of steroid hormones and PRL in cells of a high concentration PRL after interference of *MAPK12*; (**D**): The secretion of steroid hormones and PRL in cells of a high concentration of PRL after overexpression of *MAPK12*. “^#^”, “*” and “**” indicates 0.05 < *p* < 0.1, 0.01 < *p* < 0.05 and *p* < 0.01, respectively. P: GCs treated with 500 ng/mL PRL (GCs with HPCs); P-G: GCs with HPCs were infected by empty vectors of interference (negative control group for interference of *MAPK12*); P-SH: GCs with HPCs were infected by lentiviruses carrying interfered sequences of *MAPK12*; P-10: GCs with HPCs were infected by empty vectors of overexpression (negative control group for overexpression); P-10-M: GCs with HPCs were infected by lentiviruses carrying overexpressed sequences of *MAPK12*.

**Table 1 ijms-24-10269-t001:** Effects of different concentrations of PRL on apoptosis of ovine ovarian granulosa cells.

Item	PRL Supplemented, ng/mL					SEM	*p*-Value
	0	4	20	100	500		
Apoptotic rate, %	1.60 ± 0.055 ^c^	1.93 ± 0.076 ^bc^	1.56 ± 0.399 ^c^	2.07 ± 0.055 ^b^	4.01 ± 0.220 ^a^	0.009	0.023

The different lowercase letters indicate significant differences (*p* < 0.05).

**Table 2 ijms-24-10269-t002:** The information of raw data filtering.

Sample	Raw Reads	Clean Reads	Error Rate	Q20	Q30	GC pct
C1	47,452,474	46,017,302	0.03	97.28	92.75	51.02
C2	50,424,042	49,060,726	0.03	97.76	93.72	52.09
C3	45,485,418	44,286,592	0.03	97.58	93.3	51.09
C4	46,667,212	45,237,462	0.03	97.83	93.94	52.97
P1	43,582,446	42,289,948	0.03	97.69	93.62	52.82
P2	46,354,432	45,081,360	0.03	97.8	93.86	52.74
P3	48,253,124	46,923,820	0.03	97.45	93.04	52.50
P4	45,705,810	44,655,822	0.03	97.64	93.4	52.45

**Table 3 ijms-24-10269-t003:** Differentially expressed genes induced by high PRL.

Genes	Gene ID	log2 Fold Change	*p*-Value
*HSD3B1*	ENSOARG00000020402	1.32452	1.01 × 10^−16^
*STAR*	ENSOARG00000001269	1.48462	4.89 × 10^−8^
*CDC20*	ENSOARG00000020542	−1.51346	0.00037
*PLCE1*	ENSOARG00000003798	1.10614	1.35 × 10^−7^
*GRIA3*	ENSOARG00000013491	1.85560	0.00034
*IGSF10*	ENSOARG00000003977	−3.29102	0.00053
*CACNAIC*	ENSOARG00000013089	1.29897	0.00048
*RGS2*	ENSOARG00000009653	1.13720	3.20 × 10^−7^
*MAPK12*	ENSOARG00000019799	−1.18458	6.80 × 10^−16^
*CTSK*	ENSOARG00000020869	−1.39051	0.00014
*NFATC4*	ENSOARG00000019084	−2.02030	3.67 × 10^−6^

**Table 4 ijms-24-10269-t004:** Primer sequences of sgRNA and ShRNA.

Gene	Sequence 5′-3′	Accession Number
*L-PRLR*-sgRNA1	F:Caccg CAAATCCTCGCAGTCAGAAG R:Aaac CTTCTGACTGCGAGGATTTG c	O46561-1
*L-PRLR*-sgRNA2	F:Caccg CTTTGGAGGGGTGTGGCATC R:Aaac GATGCCACACCCCTCCAAAG c	O46561-1
*L-PRLR*-sgRNA3	F:Caccg TTTGCTGATGGAATTCATAG R:Aaac CTATGAATTCCATCAGCAAA c	O46561-1
*S-PRLR*-sgRNA1	F:Caccg CTTATTAAATGTCGGTCTCC R:Aaac GGAGACCGACATTTAATAAG c	O46561-2
*S-PRLR*-sgRNA2	F:Cacc GCGGTAAGTCAGTGTGTAAT R:Aaac ATTACACACTGACTTACCGC	O46561-2
*S-PRLR*-sgRNA3	F:Cacc GGAAACGTTCACCTGCTGGT R:Aaac ACCAGCAGGTGAACGTTTCC	O46561-2
*MAPK12*-ShRNA1	ACCGGCGTCATCCATAGGGACTTGACTCGAGTCAAGTCCCTATGGATGACGCTTTT	XM027968254.2
*MAPK12*-ShRNA2	ACCGGGACTGTGAGCTGAAGATTCTCTCGAGAGAATCTTCAGCTCACAGTCCTTTT	XM027968254.2
*MAPK12*-ShRNA3	ACCGGGAAGCGTGTCACATATAAAGCTCGAGCTTTATATGTGACACGCTTCCTTTT	XM027968254.2

**Table 5 ijms-24-10269-t005:** Primer sequences.

Gene	Sequence 5′-3′	Size (bp)	Tm (°C)	Accession Number
*StAR*	F:ATTCAGGAGGCAAAGAGCAGC R:TCGGGTAAGGAAAATGGGTCA	270	60	XM015094520.2
*HSD13B1*	F:CAGTCTATGTTGGCAATGTGGC R:CGGTTGAAGCAGGGGTGGTAT	283	60	NM001135932.1
*CYP11A1*	F:GTTTCGCTTTGCCTTTGAGTC R:ACAGTTCTGGAGGGAGGTTGA	120	60	NM001093789.1
*ADCY3*	F:GCAACATCCAGGTGGTGGA R:TGGTCACATGTCCTTTCCCG	282	60	XM_027966562.2
*MAPK12*	F:GCAGGCAGACAGCGAGAT R:GGTCAGGACGGAGGCAAA	307	62	XM027968254.2
*Bax*	F:TGCTCACTGCCTCACTCACC R:CCCAAGACCACTCCTCCCTA	179	60	XM027978592.1
*Bcl-2*	F:GATGACCGAGTACCTGAACCG R:GACAGCCAGGAGAAATCAAACA	120	60	XM012103831.3
*Caspase3*	F:GCTACAAGGTCCGTTATGCC R:GATGCTGCCGTATTCGTTCTC	128	60	XM015104559.2
*L-PRLR*	F:CCCCTTGTTCTCTGCTAAACCC R: CTATCCGTCACCCGAGACACC	129	60	O46561-1
*S-PRLR*	F:ACAGTAAGCGCCATCAACCA R:CTGGCTTGCATCGAATCTGC	328	60	O46561-2
*GAPDH*	F:GGTCGGAGTGAACGGATTTG R:CTTGACTGTGCCGTGGAACTT	222	60	NM001190390.1

## Data Availability

Some or all datasets generated during and analyzed during the current study are not publicly available but are available from the corresponding author (zhangyingjie@hebau.edu.cn) on reasonable request.

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
