# Peer review of "Prolactin Regulates Ovine Ovarian Granulosa Cell Apoptosis by Affecting the Expression of MAPK12 Gene"

_ijms, 2023, doi:10.3390/ijms241210269_

Round 1

Reviewer 1 Report

Introduction.

Please reword the objectives of the study to make the meaning more clear.

M & M

Please include some basic information about the reproductive physiology of Hu sheep.

Please explain why you chose that breed.

4.1. Please provide detailed analysis of feed given to animals.

4.2. Please provide criteria for allocation of animals into groups.

Please include a table with the experimental design.

Please provide names of hormone products used and their manufacturers.

Table 4. Please provide all the details of the PCRs, not just the primers.

The discussion fails to convince the reader and fails to convey a message regarding a clinical significance for the study.

The number of follicles gradu- 21 ally decreased with increasing PRL concentration.

This has been known since the 1960s….. There is little novelty in this study.

Moderate editing of English language.

Reviewer 2 Report

REVIEW

for the journal IJMS (ISSN 1422-0067)

Article

“Prolactin regulates ovine ovarian granulosa cells apoptosis by affecting the expression of MAPK12 gene

Manuscript ID: ijms-2368376

Authors:  Ruochen Yang, Chunhui Duan, Shuo Zhang, Yunqin Liu, Yingjie Zhang

The endocrine polypeptide hormone prolactin (PRL) plays an important role in lactation and maternal behavior, regulates reproductive processes and affects cell apoptosis. However, its mechanism remains unclear. Given this, the results of the authors have theoretical and practical value and an element of scientific novelty, but I would like to make a few comments that could improve the quality of this interesting article.

1)    Line 579. "The results are expressed as mean±SEM".

 The abbreviation "SEM" should be fully explained to the reader.

2)    Line 577: "Statistical analysis was performed using SPSS software (ver. 22.0, IBM Corp.). Line 588: "GraphPad Prism 9.0 software and R language (R-3.5.2) “ggplot2” package were used ...". Lines 578, 579 "The regression analysis between PRL and follicles-counts was analyzed using the “lm” package in R [64 ]".

My question: What made you choose such a variety of statistical packages?

3)    How reference 64 in the bibliography (64. Ren E, Chen X, Yu S, Xu J, Su Y, Zhu W. Transcriptomic and metabolomic responses induced in the livers of growing pigs by a short-term intravenous infusion of sodium butyrate. Animal 2018; 12: 2318-2326) is related to the statement "...the "lm" package in R [64]." (line 578).

4)     Line 65. "The correlation between PRL and the number of follicles were shown in Figure 1C". In the statistical analysis section (Lines 576 - 589), the authors did not mention that they used the correlation analysis method.

5)    Lines 64 - 68. In my opinion, the authors should comment on the obtained regression coefficients.

6)    The quality of Figure 1C, Figure 4, and Figure 5F needs to be improved as the information presented is difficult to read.

7)    Authors focused on the effect of PRL on apoptosis and possible mechanisms in ovarian GCs (ovarian granulosa cells) and they claim that this will provide a basis for future applications of PRL in ruminant reproduction. It is not clear to me, what are the practical proposals of the authors for sheep breeders and their possible contribution to future research, solving the problem of improving the reproduction of sheep?

Sincerely, reviewer.

Round 2

Reviewer 1 Report

The manuscript has been improved.
However, the discussion still requires to be extended and to become more strong going into greater depth into the topic and to better explain the findings of the present study.

Moderate editing of English language required.
